# The Kynurenine Pathway is Differentially Activated in Children with Lyme Disease and Tick-Borne Encephalitis

**DOI:** 10.3390/microorganisms9020322

**Published:** 2021-02-04

**Authors:** Ronny Wickström, Åsa Fowler, Michel Goiny, Vincent Millischer, Sofia Ygberg, Lilly Schwieler

**Affiliations:** 1Neuropediatric Unit, Department for Women’s and Children’s Health, Karolinska Institutet, 171 77 Stockholm, Sweden; ronny.wickstrom@ki.se (R.W.); sofia.ygberg@sll.se (S.Y.); 2Division of Paediatrics, Department of Clinical Science, Intervention and Technology, Karolinska Institutet, 141 52 Stockholm, Sweden; asa.fowler@sll.se; 3Department of Physiology & Pharmacology, Karolinska Institutet, 171 77 Stockholm, Sweden; Goiny.Michel@ki.se; 4Department of Psychiatry and Psychotherapy, Medical University of Vienna, 1090 Vienna, Austria; vincent.millischer@meduniwien.ac.at; 5Department of Molecular Medicine and Surgery (MMK), Karolinska Institutet, 171 77 Stockholm, Sweden

**Keywords:** tryptophan, NMDA hypofunction, cognitive symptoms, immune system

## Abstract

In children, tick-borne encephalitis and neuroborreliosis are common infections affecting the central nervous system. As inflammatory pathways including cytokine expression are activated in these children and appear to be of importance for outcome, we hypothesized that induction of the kynurenine pathway may be part of the pathophysiological mechanism. Inflammatory biomarkers were analyzed in cerebrospinal fluid from 22 children with tick-borne encephalitis (TBE), 34 children with neuroborreliosis (NB) and 6 children with no central nervous system infection. Cerebrospinal fluid levels of kynurenine and kynurenic acid were increased in children with neuroborreliosis compared to the comparison group. A correlation was seen between expression of several cerebrospinal fluid cytokines and levels of kynurenine and kynurenic acid in children with neuroborreliosis but not in children with tick-borne encephalitis. These findings demonstrate a strong induction of the kynurenine pathway in children with neuroborreliosis which differs from that seen in children with tick-borne encephalitis. The importance of brain kynurenic acid (KYNA) in both immune modulation and neurotransmission raises the possibility that abnormal levels of the compound in neuroborreliosis might be of importance for the pathophysiology of the disease. Drugs targeting the enzymes of this pathway may open the venue for novel therapeutic interventions.

## 1. Introduction

In children, tick-borne encephalitis (TBE) and neuroborreliosis (NB) are among the most common infections affecting the central nervous system (CNS) [1,2]. The reported incidence for TBE in Swedish children is around 3/100,000 [3], whereas the incidence for NB is substantially higher at 28/100,000 [4]. Although fundamentally different in etiological agents and thus appropriate treatment, TBE and NB may be similar in their clinical course. The clinical picture in children differs however, from that seen in adults and is often vague with unspecific symptoms which may delay diagnosis [5].

The pathophysiological mechanisms underlying the differences seen in outcome (varying from mild to severe) are not known, but several studies suggest that either systemic or local production of various cytokines might play a pathophysiologic role [6,7,8]. A recent study from our group confirms a strong immune activation in both children with TBE and NB and that the ratio between interleukin (IL)-6 and IL-10 is significantly different between the two types of tick-borne infections [9]. Such differences may be of clinical importance for disease severity and could serve as both biomarkers and possible targets for future therapy.

It is well known that pro-inflammatory cytokines induce tryptophan degradation via a cascade of enzymatic steps known as the “kynurenine pathway” [10] (see Figure 1). Several studies demonstrate that induction of the kynurenine pathway during infection represents a fine-tuned system for regulating both microorganisms as well as directly interfering with the immune system. Recent evidence indicates that kynurenine acts as a high-affinity ligand of the aryl hydrocarbon receptor and stimulates immunosuppressive properties [11]. In addition, kynurenine triggers regulatory T cell development [12] and some of the other tryptophan metabolites such as 3-hydroxyanthranilic and quinolinic acid (QUIN) trigger apoptosis in Th1 cells [13]. Furthermore, during infection, the kynurenine pathway suppresses the replication of microorganisms by depleting the intracellular sources of tryptophan [10].

In addition to being an essential regulator of the inherent and adaptive immune response this pathway is also responsible for the biosynthesis of neuroactive compounds, such as kynurenic acid (KYNA). This compound is synthesized in the brain by astrocytes [14] and is an antagonist at both the *N*-methyl-*D*-aspartate receptor glycine site [15] and the α7 nicotinic acetylcholine (α7nACh) receptors [16]. In particular, attenuated cholinergic and glutamatergic neurotransmission have been widely implicated in impaired cognitive functioning. Indeed, in patients with bipolar disorder increased levels of cerebrospinal fluid (CSF) KYNA are associated with impaired cognitive flexibility in terms of decreased set-shifting ability [17]. The kynurenine pathway has also been found induced in patients during infections such as herpes simplex virus type 1 encephalitis [18], NB [19], [20] and tick-borne encephalitis [21]. However, all those studies have been performed in adult patients and there are very few studies of the kynurenine pathway during infection in children.

This study provides the first comprehensive analysis of kynurenine and KYNA concentrations in CSF from children suffering from TBE and NB. We further explored if disease related changes in the kynurenine pathway correlated with CSF levels of cytokines. The results provide first evidence that there is a major induction of the kynurenine pathway in children with NB that strongly correlates to inflammatory cytokines.

## 2. Materials and Methods

### 2.1. Subjects

Inflammatory biomarkers were measured in CSF from 22 children (8 females, 14 males) with TBE (median age 142 months; interquartile range (IQR) 116–166)) and from 34 children (14 female, 20 male) with NB (median age 108 months; IQR 95–127). Children included in the study were diagnosed with TBE or NB during the period 2004–2010 in Stockholm, Sweden. Patients with TBE were retrospectively identified and retrieved from the local Swedish Institute for Infectious Disease Control and patients with identified NB were collected from the Microbiological laboratory at the Karolinska University Hospital. All CSF-samples had undergone routine virologic tests for Herpes simplex virus 1 and 2, varicella zoster virus and enterovirus with polymerase chain reaction. The CSF and serum had also tested for antibodies against Borrelia Burgdorferi in all patients with TBE and NB. Specific TBE IgM/IgG and Borrelia IgM/IgG antibodies were determined by enzyme immunoassay at the Department of Microbiology at Karolinska University Hospital according to accredited procedures. For the diagnosis of TBE, specific TBE IgM antibodies with or without TBE IgG antibodies in serum in an individual with symptoms of CNS infection, and pleocytosis (>5 × 106 (white blood cells)/L) in CSF were required. Symptoms of CNS-infection were defined as meningitis (headache, nuchal rigidity, nausea/vomiting) or encephalitis (altered mental status, focal neurologic signs, seizures, or electroencephalogram changes compatible with encephalitis. For the diagnosis of NB, a positive antibody index for CSF/serum IgM was requested together with pleocytosis (>5 × 106 (white blood cells)/L) in CSF. None of the children had a double infection with both TBE and Borrelia. Lumbar puncture was performed in the acute phase of the illness in all children. The samples had been stored at −70 C until analysis.

Six children, 4 females and 2 males (mean age 21 months; range 15–89 months) served as a comparison group of children. These children were suffering from symptoms suggestive of CNS infection and underwent a lumbar puncture in the acute phase. In all six children, CNS-infection could be ruled out and other final diagnoses were made. CSF samples from these patients did not show pleocytosis (<6 × 106 white blood cells/L).

### 2.2. Ethics

The work described in the present study was carried out in accordance with the code of ethics of the world medical association (declaration of Helsinki) for experiments including humans: “http://www.wma.net/en/30publications/10policies/b3/”. Ethical approval was obtained from the local ethics committee in Stockholm before the start of the study.

### 2.3. Analysis of KYNA and Kynurenine

KYNA and kynurenine analysis were performed with an isocratic reversed-phase high-performance liquid chromatography (HPLC) system using a a ReproSil-Pur C18 column (4 × 150 mm, Dr Maisch GmbH, Ammerbuch, Germany) and a dual piston, high liquid delivery pump (Bischoff, Leonberg, Germany). fluorescence detector (Jasco Ltd, Hachioji City, Japan) with an excitation wavelength of 344 nm and an emission wavelength of 398 nm (18 nm bandwidth) and a UV detector (shimadzu SPD-10A, wavelenght 240 nm) and essentially as previously described [22].

Sodium acetate pH (50 mM, pH6.20 adjusted with acetic acid) and 7.0% acetonitrile were used as mobile phase and pumped through the reversed-phase column at a flow rate of 0.5 mL/min. CSF samples (30 µL) were manually injected (Rheodyne, Rhonert Park, CA, USA). Zinc acetate (0.5 M, not pH adjusted) was delivered post column by a peristaltic pump (P-500, Pharmacia, Uppsala, Sweden) with a flow rate of 10 mL/h. The retention time of KYNA was about 7 min and kynurenine 4 min. The software Datalys Azur (version 4.6.0.0;http://datalys.net) was used to calculate the signals from the fluorescence detector. The sensitivity of the system was verified by analysis of standard mixture of KYNA (0.5–30 nM) and kynurenine (5–2500 nM), which resulted in a linear standard plot. To verify the reliability of this method, some samples were analyzed in duplicate, and the mean intraindividual variation was below 5%.

### 2.4. Analysis of Cytokines

Cytokines were measured in the CSF from the diagnostic lumbar puncture of 20 children with TBE, 29 children with neuroborreliosis and 6 children without evidence of central nervous system infection, using a premade multiplex assay (Bio-Plex Pro; Life Science Bio-Rad, Solna, Sweden) for analysis of IL-1ß, IL1-Receptor, IL-4, IL-6, IL-7, IL-8, IL-10, IL-12 (p70), IL-13, IL15, IL-17, IL-18, IFN-γ, MCP-1, and IP10. The panel was chosen as it contained pro-inflammatory cytokines that were considered relevant and that were likely to be available in clinical practice. IL-1b, IL-15, IL-17 and IP10 were undetectable in more than 70% of all samples tested and therefore excluded from further analysis. A full description of the panel and cytokine results has been published previously [9]. After assessment with principal component analysis 2 subjects (one TBE and one NB) were excluded.

### 2.5. Statistical Analysis

Given the small sample size and the skewed distribution of metabolite concentrations in each group, all values are given as median with IQR. Outliers were identified with Grubb’s test. A two-sided Mann–Whitney U-test with Bonferroni correction for repeated comparisons was used to identify differences regarding CSF levels of kynurenine and KYNA between the groups (GraphPad Prism version 8.4.3 for macOS, GraphPad Software, La Jolla, CA, USA, www.graphpad.com). The alpha level of significance was set at 0.05 and with Bonferroni correction for three tests (within each metabolite) *p*-values < 0.016 were considered significant. Correlation analyses were performed using a Spearman rank correlation analysis using R programming language (R version 4.0.2) [23]. Graphs were created using the package ggplot [24]. Given the exploratory nature of the correlation analysis, the alpha level of significance was set at 0.05.

## 3. Results

### 3.1. CSF Levels of Kynurenine and KYNA

CSF concentrations of kynurenine or KYNA were not statistically significant different between children with TBE (kynurenine: 0.06 μM, IQR 0.04–0.1 μM, *n* = 21, *p* = 0.7, KYNA: 4.5 nM, IQR 2.9–9.6 nM, *n* = 21, *p* = 0.1, see Figure 2A,B) and children included in the comparison group (kynurenine 0.06 μM, IQR 0.06–0.08 μM, *n* = 5 and KYNA 3.7 nM, IQR 1.8–3.8 nM, *n* = 5, see Figure 2A,B). In contrast, kynurenine concentrations were markedly increased in children with NB (kynurenine: 0.99 μM, IQR 0.7–1.3 μM, *n* = 34, *p* < 0.0001 and KYNA: 6.6 nM, IQR 4.6–8.9 nM, *n* = 34, *p* = 0.005 see Figure 2A,B) compared to the comparison group. Kynurenine was statistically significantly increased in patients with NB compared to children with TBE (*p* < 0.0001, see Figure 2A). Kynurenine correlated positively with KYNA in children with NB (Spearman’s rho = 0.7, *n* = 34, *p* = 0.0001, see Figure 2C) but not in children with TBE (Spearman’s rho = 0.1, *n* = 21, *p* = 0.3, see Figure 2C).

### 3.2. Correlations between CSF Kynurenine or KYNA with Inflammatory Cytokines

We have previously reported the cytokine profile in this cohort of children with TBE and NB [9]. Here we investigated associations between 11 inflammatory cytokines detected in CSF with tryptophan metabolites in children with NB or TBE (see Figure 3A). We found that seven cytokines correlated to kynurenine and five to KYNA in CSF from children with NB, but no correlations were found with any of the tryptophan metabolites in CSF from children with TBE (see Figure 3A). In children with NB CSF kynurenine correlated positively with IFN-γ (Spearman’s rho = 0.6, *n* = 28, *p* = 0.00066, see Figure 3B); IL8 (Spearman’s rho = 0.58, *n* = 28, *p* = 0.0016, see Figure 3B); IL-10 (Spearman’s rho = 0.51, *n* = 28, *p* = 0.0057, see Figure 3B); IL18 (Spearman’s rho = 0.57, *n* = 28, *p* = 0.0017, see Figure 3B); IL-4 (Spearman’s rho = 0.48, *n* = 28, *p* = 0.01); IL-6 (Spearman’s rho = 0.44, *n* = 28, *p* = 0.02) and negatively with IL-12p70 (Spearman’s rho= −0.41, *n* = 28, *p* = 0.03). KYNA showed a positive correlation with IFN-γ (Spearman’s rho = 0.51, *n* = 28, *p* = 0.0057 see Figure 3C); IL-4 (Spearman’s rho = 0.38, *n* = 28, *p* = 0.05); IL-8 (Spearman’s rho = 0.46, *n* = 28, *p* = 0.014); IL-18 (Spearman’s rho = 0.42, *n* = 28, *p* = 0.026), and negatively with IL-12p70 (Spearman’s rho= −0.48, *n* = 28, *p* = 0.01).

## 4. Discussion

The present study clearly demonstrates that children with NB have a strong induction of the kynurenine pathway, with increased levels of kynurenine and KYNA during the acute phase of the disorder. A strong correlation between CSF kynurenine and KYNA further supports an overactive kynurenine pathway in in these children. To our surprise, children with TBE had CSF kynurenine levels in the range of what was seen in the comparison group and differ, in this regard, from the children with NB and also from adult patients with TBE that display a strong induction of the kynurenine pathway at the acute phase of the disease [20].

When investigating the association of cytokines with tryptophan metabolites, we found that seven of 11 cytokines associated to kynurenine and five to KYNA in CSF from children with NB, but no correlations were found between any of the tryptophan metabolites and cytokines in CSF from children with TBE. IFN-γ show a strong association to both kynurenine and KYNA in children with NB indicating its importance as an inducer of the pathway in CSF. It is well established that this pro-inflammatory cytokine increases the expression of the first limiting enzyme indoleamine 2,3-dioxygenase (IDO1) which transforms tryptophan to kynurenine [25,26,27,28] (see Figure 1) resulting in cellular tryptophan depletion. However, even though IFN-γ is highly activated in children with TBE, with individual concentrations higher than found in children with NB (see Figure 3B), it did not induce kynurenine in these children. The blunted action of IFN-γ on tryptophan metabolism in children with TBE compared to children with NB is obscure but might be explained by the need for a co-induction of the pathway by several cytokines. Interestingly, cytokines that strongly correlate with kynurenine in children with NB in the present study (IL-4, IL-6, IL-8, IL-10 and IL-18) have been shown to be of importance for the induction of IDO1 either alone for IL-6 [29] and IL-18 [30] or together with IFN-γ and IL-4 [31], IL8 [32] and IL-10, [33,34]. IL-10 could be of special interest, in this regard, since this cytokine is only increased in children with NB and not in children with TBE [9]. Furthermore, adult TBE patients, unlike infants, have elevated levels of IL-10 [35], which may explain the discrepancy in CSF levels of kynurenine between children and adults during the acute phase of TBE. However, in order to better understand the mechanism behind the variations observed between young and adult patients, further research, including both cytokine and kynurenine profiling in TBE patients, is needed.

We should also take into account the essential role of the kynurenine pathway in balancing the activation and suppression of the immune system. [36] In fact, several studies have suggested that immune system cells use bidirectional communication in which IDO activity drives the generation of IL-10 producing regulatory T-cells [37]. Induction of the kynurenine pathway would, thus, prevent an exaggerated immune response, and help to restore the homeostasis.

The strong induction of the kynurenine pathway in children with NB could also be a result from decreased expression of tryptophanyl-tRNA synthetase (WRS). This protein catalyzes the attachment of tryptophan to its cognate transfer RNA molecule, that directly will provide an accessible reservoir of tryptophan for protein synthesis [38,39]. Thus, it has been shown that in parallel to an induction of IDO1, WRS expression decreases during stimulation with IL-4 in the presence of IFN-γ in primary microglial cells [30]. In the absence of WRS more tryptophan will be available for the kynurenine pathway and might explain the high concentrations of CSF kynurenine and its correlation to IL-4 in children with NB.

Although most patients with NB are successfully treated by timely antibiotic therapy, it is broadly accepted that many patients experience treatment failure and continue to suffer long-term, debilitating symptoms, including pain, fatigue, cognitive dysfunction and other symptoms [40,41]. This is known as post-treatment lyme disease (PTLD) and the numbers of PTLD cases in the US have been estimated to be around 2 million in 2020 [42], suggesting a substantial number of patients living with significant health challenges. The number of studies of PTLD in children are relatively sparse, include a small number of subjects and show inconclusive results. Two studies reported a good long-term clinical recovery in children with confirmed NB [43,44]. However, other studies point to both neuropsychiatric and cognitive abnormalities in children with NB [45,46]. The reason why some individuals develop clinical manifestations after NB infection while others remain asymptomatic is largely unknown. Interestingly, several studies have demonstrated that altered cytokine levels and associated dysregulation of kynurenine metabolism plays an important role in the development of neuropsychiatric symptoms. Thus, in recent years studies on the role of endogenous KYNA in controlling cognitive functions have gained increased attention due to its unique ability to inhibit both glutamatergic and cholinergic receptors that are of critical importance for physiological processes underlying learning and memory [47]. Pharmacological elevation of brain KYNA or genetic ablation of the enzyme kynurenine monooxygenase (KMO) in rodents results in impaired memory and learning [48] and reduced cognitive flexibility [49]. The importance of brain KYNA in both immune modulation and neurotransmission raises the possibility that abnormal levels of the compound in NB might be of importance for the pathophysiology of the disease. Indeed, a recent study has shown that cytokine induced IDO1 correlates with pathogenic potential in human immune cells infected with Borrelia burgdorferi [50]. 

Children with TBE also suffer from sequela including cognitive symptoms [51]. A recent study from our lab shows that several CSF cytokines (IFN-gamma, IL-4, IL-6 and IL-8) are increased in children (mean age 11 years) who later developed persistent sequela (including cognitive symptoms) [52]. The mechanism important for developing sequela appears related to the grade of inflammation in CNS rather than direct neural destruction since an inverse relationship with the neuro specific enolase (NSE) and sequela were found in these TBE children. In the present study, children with TBE did not significantly differ from the comparison group or from the children with NB in CSF levels of KYNA. However, as much as 50% of all TBE children had a higher concentration of CSF KYNA than any of the children in the comparison group (see Figure 2B) and it is possible that some of these children may develop cognitive symptoms due to increased CSF levels of KYNA.

A detailed knowledge of endogenous brain kynurenine and KYNA during the course of TBE and NB infection might yield further insights into the neuroimmunological role of the compound and may also provide new pharmacological approaches for the treatment of cognitive symptoms in these children.

## 5. Strength and limitation

The limitation of the present study is the small number as well as the younger age of the children representing the comparison group. The age differences in CSF kynurenine or KYNA in young healthy children has never been tested, however in experimental studies both kynurenine and KYNA concentrations have been found higher in fetal brain [53,54] and decreased in the postnatal period and in adult age [55]. The children in the comparison group could in this regard give false high levels of kynurenine and KYNA due to their younger age but were found to be in the same range or even lower than the levels detected in children with TBE or NB. Furthermore, the comparison group of the present study showed CSF levels of both kynurenine and KYNA of the same magnitude as those found in one other study of healthy children [56]. A second limitation of the study were the lack of clinical outcome or quantitative behavioral data in children. We could therefore not investigate whether changes in CSF concentrations of kynurenine or KYNA had any prognostic value or correlated with cognitive performance or mood changes, as seen in adults with induced kynurenine pathway [17,57,58]. Moreover, the levels of cytokines and KYNA metabolites could differ in CSF depending on disease duration. However, even though the exact time from onset of disease until collection of samples was unknown for the group of NB, pleocytosis in CSF as well as positive antibody index for CSF/serum IgM indicate that children were in the acute phase of the disease. For children with TBE CSF samples were taken on median day 5 (IQR 3-10) after disease onset. The major strengths of the study are the measurement of kynurenine, KYNA and cytokines in precious CSF samples from children. Neither cytokines nor KYNA (owing to its polar structure), penetrates through the blood–brain barrier (BBB) and it is therefore not expected that plasma or serum values would reflect changes in the brain. Furthermore, correlations between CSF cytokines and kynurenine metabolites from the same CSF samples can shed light on the importance of cytokines for the induction of the kynurenine pathway.

## 6. Conclusions

As far as we are aware, this is the first study investigating the kynurenine metabolism in children with NB and TBE. Our results show a strong induction of the kynurenine pathway with increased CSF levels of kynurenine and KYNA in children with NB compared to children with TBE. The pathway is sensitive to inflammatory signaling and its metabolites have neuromodulatory properties. Inhibition of particular enzymes in this pathway has been used to improve symptomatology in many animal models of CNS disease. Successful translation of these pre-clinical efforts into real drugs may open the stage for novel therapeutic approaches to human CNS disease.

## Figures and Tables

**Figure 1 microorganisms-09-00322-f001:**
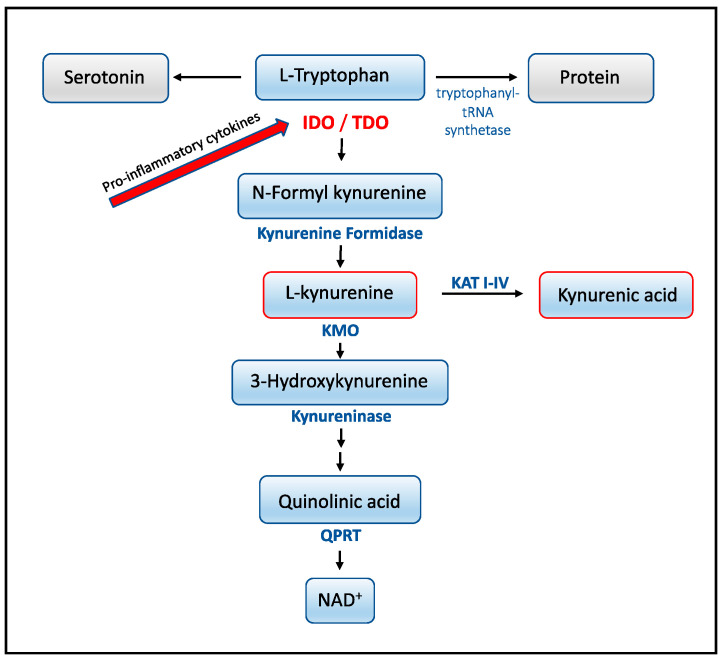
Schematic overview of the kynurenine pathway. The first and rate-limiting step is catalyzed by indoleamine 2,3-dioxygenase (IDO1 and 2) or by tryptophan 2,3-dioxygenase (TDO2) that transform tryptophan to *N*-formyl kynurenine. The second enzyme in the pathway kynurenine formamidase produces *L*-kynurenine that could enter different possible branches, depending on cell-type or environmental context, to form various metabolites such as kynurenic acid (KYNA) by kynurenine aminotransferases (KAT I-IV) or quinolinic acid via kynurenine mono oxidase (KMO). Pathways represented by interrupted arrows involve several metabolites and enzymatic reactions.

**Figure 2 microorganisms-09-00322-f002:**
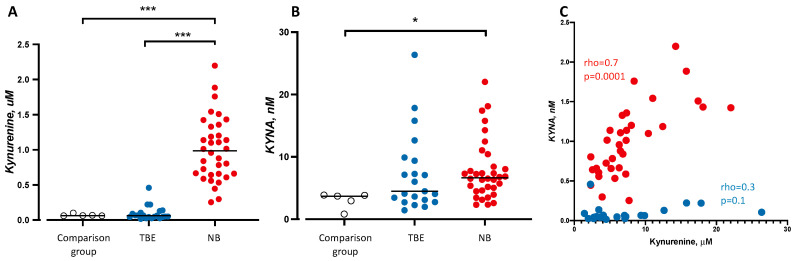
Kynurenine, (**A**) kynurenic acid (**B**, KYNA) and the correlation between KYNA and Kynurenine (**C**) in human cerebrospinal fluid (CSF). Each point represents the concentration of a single CSF sample in units of μM (**A**: kynurenine) or nM (**B**: KYNA) together with the median for children in the comparison group (white circles, *n* = 5), children with tick-borne encephalitis (TBE, blue circles, *n* = 21), and children with neuroborreliosis (NB, red circles, *n* = 34). Two outliers for each metabolite were identified with Grubb’s test. Differences between groups were tested with a Mann–Whitney U-test; reported *p*-values are two-sided. Alpha threshold * *p* < 0.016; *** *p* < 0.0003 after Bonferroni correction. Correlation (**C**) between concentrations of kynurenine and KYNA in children with NB (red circles, *n* = 34) and TBE (blue circles, *n* = 21). Spearman rank correlation analysis (rho). Alpha threshold is set to *** *p* < 0.001.

**Figure 3 microorganisms-09-00322-f003:**
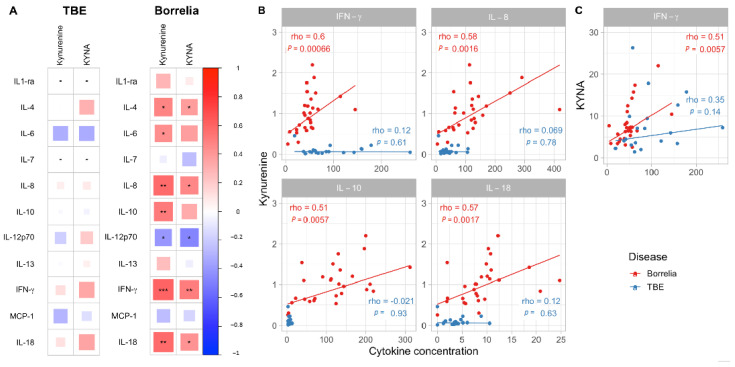
(**A**). Heatmaps of Spearman correlation coefficients between CSF tryptophan metabolites kynurenine, kynurenic acid (KYNA) and CSF cytokines in children with neuroborreliosis (NB, *n* = 28) and tick-borne encephalitis (TBE, (*n* = 22). Positive correlations (blue), Negative correlations (red). (**B**) Correlation between concentrations of kynurenine and IFN-γ, interleukin (IL)-8, IL10 and IL-18 and (**C**) Correlation between concentrations of KYNA and interferon (IFN)-γ in children with NB (red circles) and TBE (blue circles). Spearman rank correlation analysis (rho). Alpha threshold is set to *** *p* < 0.001, ** *p* < 0.01, * *p* < 0.05.

## Data Availability

Given that the whole dataset from the present study may allow individual identification, which is against the General Data Protection Regulation (GDPR), researchers need to request the data from the corresponding author.

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
