# Peer review of "The Kynurenine Pathway is Differentially Activated in Children with Lyme Disease and Tick-Borne Encephalitis"

_microorganisms, 2021, doi:10.3390/microorganisms9020322_

Round 1
Reviewer 1 Report
An interesting paper on an important subject. Clearly described methodology and results.
A suggestion : include an original paper on CSF kynurenine in adults with NB into the references,
Fuhs D, Dotevall L, Hagberg L,Werner E and Wächter H: Kynurenine in cerebrospinal fluid of patients with Lyme neuroborreliosis. Immun Infect Dis 1991:1: 271-274.
Author Response
Reviewer 1
An interesting paper on an important subject. Clearly described methodology and results.
A suggestion : include an original paper on CSF kynurenine in adults with NB into the references,
Fuhs D, Dotevall L, Hagberg L,Werner E and Wächter H: Kynurenine in cerebrospinal fluid of patients with Lyme neuroborreliosis. Immun Infect Dis 1991:1: 271-274.
Thanks for suggested reference, it is added to the manuscript:
Introduction,Line 87 (reference 20).
References, Line 437-439.
Reviewer 2 Report
Wickström et al., measured the cerebrospinal fluid levels of kynurenine and kynurenic acid and certain citokines in samples of children with neuroborreliosis, tick-borne encephalitis, and controls. They could detect elevated level of kynurenine and kynurenic acid and certain citokins in NB samples but not in TBE cases and controls. The applied methods are appropriate, the results are presented, explained correctly with good English.
In some respect the results are not suprising as clinical course of Lyme disease and TBE and their agents are so different. For me the most interesting part was that kynurenine and kynurenic acid levels were low in children with TBE, but higher in adults (Holtze et al., 2012). TBE does not cause serious clinical signs in youngs but could be much severe in adults and olds. It is not possible, that activated kynurenine pathway is only result of the presence of bacteria and virus in CNS?
I really missed three things in this paper.
1., A simple PCR of the CNS fluid samples for the presence of the virus and the bacteria should have been necessary. Were there at all the agents in the brain in these patients? I really do not understand why the authors not started this work with a direct virus/bacteria detection from the CNS samples by PCRs.
2., And some details about the clinical course of TBE and LB in the youngs, whose CNS fluid the authors used. How were they diagnosed as TBE patients (methods)?
I have never heard of clinically ill TBE patients around (or younger) 10 years of age. The authors wrote, that the source of their samples, the children „were diagnosed with TBE or NB during 2004-2010 „ How were they diagnosed? tests? Clinical sgns? Presence of the pathogens?
3., Discussion. What the authors think, why kynurenine and KYNA leves are higher in adult patients than in children with TBE? Some views/ideas could have been written.
As this work help in better understanding the immune response to two important tick-borne zoonoses, I suggest its publication, but I ask the authors, in their revised manuscrip give some detailed data about the clinical course and diagnosis of the children whose samples they worked from and check their CNS fluid samples for the presence of TBEV and LB genomes.
row 210 ’in in these children’
Author Response
- A simple PCR of the CNS fluid samples for the presence of the virus and the bacteria should have been necessary. Were there at all the agents in the brain in these patients? I really do not understand why the authors not started this work with a direct virus/bacteria detection from the CNS samples by PCRs.
Thank you for your comment. All CSF samples were collected retrospectively from a clinical setting were analysis of viruses and bacteria had already been made. In our region, a test battery with PCR for viral detection of HSV1/2, VZV and enterovirus in CSF is a routine testing when CNS-infection is suspected, as well as bacterial culture. These testes had been done in all patients before our retrieval of the samples. Furthermore, all patients were also routinely tested for intrathecal and serum antibodies against Borrelia burgdorferi. PCR-testing for TBEV was not available at a clinical setting at the time of the study and is only analyzed at specialized laboratories, even today. To our knowledge, PCR-tests for detection of Borrelia burgdorferi is not in clinical use. Due to small sample-volume of the retrieved CSF samples, a new PCR-testing for viruses/bacteria could not be made and was not considered necessary since it already had been performed. The fact that screening of the CSF-samples for viruses and bacteria had already been performed when samples were collected from the patient in the clinical setting have been clarified in the method section.
Method section, subjects, line 103:All CSF-samples had undergone routine virologic tests for Herpes simplex virus 1 and 2, varicella zoster virus and enterovirus with polymerase chain reaction. The CSF and serum had also tested for antibodies against Borrelia Burgdorferi in all patients with TBE and NB. Specific TBE IgM/IgG and Borrelia IgM/IgG antibodies were determined by enzyme immunoassay at the Department of Microbiology at Karolinska University Hospital according to accredited procedures.
- And some details about the clinical course of TBE and LB in the youngs, whose CNS fluid the authors used. How were they diagnosed as TBE patients (methods)?
I have never heard of clinically ill TBE patients around (or younger) 10 years of age. The authors wrote, that the source of their samples, the children „were diagnosed with TBE or NB during 2004-2010 „ How were they diagnosed? tests? Clinical sgns? Presence of the pathogens?
Thanks for pointing out the limitation in the presentation of diagnosis criteria. We have now included a more detailed description of the diagnosis in the Material and methods section “subjects”.
Method section, subjects, line 104-115:All CSF-samples had undergone routine virologic tests for Herpes simplex virus 1 and 2, varicella zoster virus and enterovirus with polymerase chain reaction. The CSF and serum had also tested for antibodies against Borrelia Burgdorferi in all patients with TBE and NB. Specific TBE IgM/IgG and Borrelia IgM/IgG antibodies were determined by enzyme immunoassay at the Department of Microbiology at Karolinska University Hospital according to accredited procedures. For the diagnosis of TBE, specific TBE IgM antibodies with or without TBE IgG antibodies in serum in an individual with symptoms of CNS infection, and pleocytosis (>5x 106 (WBC)/L) in CSF were required. Symptoms of CNS-infection were defined as meningitis (headache, nuchal rigidity, nausea/vomiting) or encephalitis (altered mental status, focal neurologic signs, seizures, or EEG changes compatible with encephalitis). For the diagnosis of NB, a positive antibody index for CSF/serum IgM was requested together with pleocytosis (>5x 106 (WBC)/L) in CSF.
3. Discussion. What the authors think, why kynurenine and KYNA leves are higher in adult patients than in children with TBE? Some views/ideas could have been written.
The reviewer is correct and the difference in the induction of the kynurenine pathway between adults and children with TBE is very interesting. A direct comparison between TBE adults and children is, however challenging since a detailed cytokine profile and its association to kynurenine metabolites in these patients is very limited in the literature. We have though expanded the discussion about IL-10 and its importance for induction of the kynurenine pathway that might shed some light on the difference between the induction of the kynurenine pathway in adult and children with TBE.
Discussion, line 235-241:
IL-10 could be of special interest since this cytokine is only increased in children with NB and not in children with TBE [9]. Adult TBE patients, unlike infants, have elevated levels of IL-10 [35], which may explain the discrepancy in kynurenine metabolism during the acute phase of TBE. However, in order to better understand the mechanism behind the variations observed between young and adult TBE patients, further research, including both cytokine and kynurenine profiling in TBE patients, are needed.
Reviewer 3 Report
This study assessed the levels of kynurenin and kynurenic acid (KYNA) in CSF of children with LNB or TBE and correlated them with cytokine and chemokine levels. The manuscript is generally well written and organized. I have primarily minor comments for improvement. Specific suggestions are provided below.
1) The authors should present the cytokine levels as a standalone figure. I think this will further showcase the differences between LNB and TBE. A very interesting finding is that children with TBE have high levels of IFNy, but very low levels of IL-10 (a key anti-inflammatory cytokine). Presenting the cytokine data as a separate Figure would highlight this difference.
2) One of potential disconnects between kynurenin pathway and cytokines in LNB vs TBE could be a difference in duration of disease at the time CSF was obtained for each disease. If this information is available, please include it along with the discussion.
3) The majority of the discussion currently focuses on the potential for inflammatory and pathogenic effects of kynurenin/KYNA. However, the data also suggests the possibility that the kynurenin levels may actually be increased in an effort to suppress the heightened levels of proinflammatory cytokines in LNB. This idea is also supported by heightened IL-10 levels in LNB. Do IL-10 levels correlate with IFNy levels? It is possible that patients with TBE have heightened IFNy responses because of a defect in a regulatory IL-10 / kynurenine response. A discussion of this is warranted.
4) The authors imply that abnormal levels of KYNA could contribute to pathophysiology particularly related to post-treatment symptoms in LNB. However, TBE is also associated with persistent complications (post-encephalitic syndrome). Since KYNA levels are similar in LNB and TBE, is it possible that elevated KYNA levels may impact the neurocognitive deficits in both disorders, independent of cytokine levels? Conversely, do the authors think that post-infectious complications in TBE and LNB are distinct and thus involve different mechanisms? Ultimately, the answers to these questions will require the studies of patients with detailed clinical information on the course and outcome of each condition. However, since this is the first such study, it would help to speculate about this in the discussion.
Author Response
Reviewer 3
This study assessed the levels of kynurenin and kynurenic acid (KYNA) in CSF of children with LNB or TBE and correlated them with cytokine and chemokine levels. The manuscript is generally well written and organized. I have primarily minor comments for improvement. Specific suggestions are provided below.
1)The authors should present the cytokine levels as a standalone figure. I think this will further showcase the differences between LNB and TBE. A very interesting finding is that children with TBE have high levels of IFNy, but very low levels of IL-10 (a key anti-inflammatory cytokine). Presenting the cytokine data as a separate Figure would highlight this difference.
We agree with the reviewer that a separate figure of the cytokines alone would be of value. However, all cytokine data has already been published and maynotbe published again due to rules of plagiarism. To highlight this, we have the reference with cytokine data (Ygberg et al., 2020) in the initial sentence of the results section concerning correlations between kynurenine /KYNA and cytokines.
2) One of potential disconnects between kynurenin pathway and cytokines in LNB vs TBE could be a difference in duration of disease at the time CSF was obtained for each disease. If this information is available, please include it along with the discussion.
We agree with the reviewer that the timepoint of the analysis of CSF may affect the levels of cytokines, kynurenine and KYNA measured in the CSF. In the TBE-cases, the CSF was obtained during the acute phase of the illness in all cases, with a median of 5 days (IQR 3-10). However, in NB cases no data regarding the timepoint when the sample was collected were available. For the diagnosis of NB, pleocytosis in CSF and a positive antibody index for CSF/serum IgM was requested, thus indicating that all samples were collected in the acute phase. A paragraph regarding this has been added to the discussion section.
Discussion, line 306-311: Moreover, the levels of cytokines and KYNA metabolites could differ in CSF depending on disease duration. However, even though the exact time from onset of disease until collection of samples was unknown for the group of NB, pleocytosis in CSF as well as positive antibody index for CSF/serum IgM indicate that children were in the acute phase of the disease. For children with TBE CSF samples were taken on median day 5 (IQR 3-10) after disease onset.
3) The majority of the discussion currently focuses on the potential for inflammatory and pathogenic effects of kynurenin/KYNA. However, the data also suggests the possibility that the kynurenin levels may actually be increased in an effort to suppress the heightened levels of proinflammatory cytokines in LNB. This idea is also supported by heightened IL-10 levels in LNB. Do IL-10 levels correlate with IFNy levels? It is possible that patients with TBE have heightened IFNy responses because of a defect in a regulatory IL-10 / kynurenine response. A discussion of this is warranted.
Indeed, the essential role of the kynurenine pathway in suppression of the immune system was missing in the manuscript and have now been addedintothe Discussion.
Discussion, line 242-246:We should also take into account the essential role of the kynurenine pathway in balancing the activation and suppression of the immune system [36] In fact, several studies have suggested that immune system cells use bidirectional communication in which IDO activity drives the generation of regulatory T-cells producing IL-10 [37]. Induction of the kynurenine pathway would, thus, preventanexaggerated immune response, andhelp to restore the homeostasis.
4) The authors imply that abnormal levels of KYNA could contribute to pathophysiology particularly related to post-treatment symptoms in LNB. However, TBE is also associated with persistent complications (post-encephalitic syndrome). Since KYNA levels are similar in LNB and TBE, is it possible that elevated KYNA levels may impact the neurocognitive deficits in both disorders, independent of cytokine levels? Conversely, do the authors think that post-infectious complications in TBE and LNB are distinct and thus involve different mechanisms? Ultimately, the answers to these questions will require the studies of patients with detailed clinical information on the course and outcome of each condition. However, since this is the first such study, it would help to speculate about this in the discussion.
Thanks for a very important question, we have now included a section in the end of the Discussion that include the possibility that some of the children with TBE may develop cognitive symptoms due to increased CSF levels of KYNA.
Discussion, Line: 277-286
Children with TBE also suffer from sequela including cognitive symptoms [51]. A recent study from our lab shows that several CSF cytokines (IFN-gamma, IL-4, IL-6 and IL-8) are increased in children (mean age 11 years) who later developed persistent sequela (including cognitive symptoms) [52]. The mechanism important for developing sequela appears related to the grade of inflammation in CNS rather than direct neural destruction since an inverse relationship with the neuro specific enolase (NSE) and sequela were found in these TBE children. In the present study, children with TBE did not significantly differ from the comparison group or from the children with NB in CSF levels of KYNA. However, as much as 50% of all TBE children had higher concentration of CSF KYNA than any of the children in the comparison group (see figure 2B) and it is possible that some of these children may develop cognitive symptoms due to increased CSF levels of KYNA.
Round 2
Reviewer 2 Report
line 231 in in the children (STILL)